# Evolutionary Stochastic Gradient Descent for Optimization of Deep Neural Networks

**Xiaodong Cui,  Wei Zhang,  Zoltán Tüske  and  Michael Picheny**
IBM Research AI
IBM T. J. Watson Research Center
Yorktown Heights, NY 10598, USA
`{cuix, weiz, picheny}@us.ibm.com, {Zoltan.Tuske}@ibm.com`

## Abstract

We propose a population-based Evolutionary Stochastic Gradient Descent (ESGD) framework for optimizing deep neural networks. ESGD combines SGD and gradient-free evolutionary algorithms as complementary algorithms in one framework in which the optimization alternates between the SGD step and evolution step to improve the average fitness of the population. With a back-off strategy in the SGD step and an elitist strategy in the evolution step, it guarantees that the best fitness in the population will never degrade. In addition, individuals in the population optimized with various SGD-based optimizers using distinct hyper-parameters in the SGD step are considered as competing species in a coevolution setting such that the complementarity of the optimizers is also taken into account. The effectiveness of ESGD is demonstrated across multiple applications including speech recognition, image recognition and language modeling, using networks with a variety of deep architectures.

## 1   Introduction

Stochastic gradient descent (SGD) is the dominant technique in deep neural network optimization [1]. Over the years, a wide variety of SGD-based algorithms have been developed [2, 3, 4, 5]. SGD algorithms have proved to be effective in optimization of large-scale deep learning models. Meanwhile, gradient-free evolutionary algorithms (EA) [6, 7, 8, 9, 10] also have been used in various applications. They represent another family of so-called black-box optimization techniques which are well suited for some non-linear, non-convex or non-smooth optimization problems. Biologically inspired, population-based EA make no assumptions on the optimization landscape. The population evolves based on genetic variation and selection towards better solutions of the problems of interest. In deep learning applications, EA such as genetic algorithms (GA), evolutionary strategies (ES) and neuroevolution have been used for optimizing neural network architectures [11, 12, 13, 14, 15] and tuning hyper-parameters [16, 17]. Applying EA to the direct optimization of deep neural networks is less common. In [18], a simple EA is shown to be competitive to SGD when optimizing a small neural network (around 1,000 parameters). However, competitive performance on a state-of-the-art deep neural network with complex architectures and more parameters is yet to be seen.

The complementarity between SGD and EA is worth investigating. While SGD optimizes objective functions based on their gradient or curvature information, gradient-free EA sometimes are advantageous when dealing with complex and poorly-understood optimization landscape. Furthermore, EA are population-based so computation is intrinsically parallel. Hence, implementation is very suitable for large-scale distributed optimization. In this paper we propose Evolutionary Stochastic Gradient Descent (ESGD) – a framework to combine the merits of SGD and EA by treating them as complementary optimization techniques.

Given an optimization problem, ESGD works with a population of candidate solutions as individuals. Each individual represents a set of model parameters to be optimized by an optimizer (e.g. conventional SGD, Nesterov accelerated SGD or ADAM) with a distinct set of hyper-parameters (e.g. learning rate and momentum). Optimization is carried out by alternating SGD and EA in a stage-wise manner in each generation of the evolution. Following the EA terminology [6, 19, 20], consider each individual in the population as a "species". Over the course of any single generation, each species evolves independently in the SGD step, and then interacts with each other in the EA step. This has the effect of producing more promising candidate solutions for the next generation, which is coevolution in a broad sense. Therefore, ESGD not only integrates EA and SGD as complementary optimization strategies but also makes use of complementary optimizers under this coevolution mechanism. We evaluated ESGD in a variety of tasks. Experimental results showed the effectiveness of ESGD across all of these tasks in improving performance.

## 2   Related Work

The proposed ESGD is pertinent to neuroevolution [21, 22] which consists of a broad family of techniques to evolve neural networks based on EA. A large amount of work in this domain is devoted to optimizing the networks with respect to their architectures and hyper-parameters [11, 12, 15, 16, 23, 24]. Recently remarkable progress has been made in reinforcement learning (RL) using ES [22, 25, 26, 27, 28]. In the reported work, EA is utilized as an alternative approach to SGD and is able to compete with state-of-the-art SGD-based performance in RL with deep architectures. It shows that EA works surprisingly well in RL where only imperfect gradient is available with respect to the final performance. In our work, rather than treating EA as an alternative optimization paradigm to replace SGD, the proposed ESGD attempts to integrate the two as complementary paradigms to optimize the parameters of networks.

The ESGD proposed in this paper carries out population-based optimization which deals with a set of models simultaneously. Many of the neuroevolution approaches also belong to this category. Recently population-based techniques have also been applied to optimize neural networks with deep architectures, most notably population-based training (PBT) in [17]. Although both ESGD and PBT are population-based optimization strategies whose motivations are similar in spirit, there are clear differences between the two. While evolution is only used for optimizing the hyper-parameters in PBT, ESGD treats EA and SGD as complementary optimizers to directly optimize model parameters and only indirectly optimize hyper-parameters. We investigate ESGD in the conventional setting of supervised learning of deep neural networks with a fixed architecture without explicit tuning of hyper-parameters. More importantly, ESGD uses a model back-off and elitist strategy to give a theoretical guarantee that the best model in the population will never degrade.

The idea of coevolution is used in the design of ESGD where candidates under different optimizers can be considered as competing species. Coevolution has been widely employed for improved neuroevolution [19, 20, 29, 30] but in cooperative coevolution schemes species typically represent a subcomponent of a solution in order to decompose difficult high-dimensional problems. In ESGD, the coevolution is carried out on competing optimizers to take advantage of their complementarity.

## 3   Evolutionary SGD

### 3.1   Problem Formulation

Consider the supervised learning problem. Suppose $\mathcal{X} \subseteq \mathbb{R}^{d_x}$ is the input space and $\mathcal{Y} \subseteq \mathbb{R}^{d_y}$ is the output (label) space. The goal of learning is to estimate a function $h$ that maps from the input to the output

$$h(x; \theta) : \mathcal{X} \to \mathcal{Y} \tag{1}$$

where $x \in \mathcal{X}$ and $h$ comes from a family of functions parameterized by $\theta \in \mathbb{R}^d$. A loss function $\ell(h(x; \theta), y)$ is defined on $\mathcal{X} \times \mathcal{Y}$ to measure the closeness between the prediction $h(x; \theta)$ and the label $y \in \mathcal{Y}$. A risk function $R(\theta)$ for a given $\theta$ is defined as the expected loss over the underlying joint distribution $p(x, y)$:

$$R(\theta) = \mathbb{E}_{(x,y)}[\ell(h(x; \theta), y)] \tag{2}$$

We want to find a function $h(x; \theta^*)$ that minimizes this expected risk. In practice, we only have access to a set of training samples $\{(x_i, y_i)\}_{i=1}^n \in \mathcal{X} \times \mathcal{Y}$ which are independently drawn from $p(x, y)$. Accordingly, we minimize the following empirical risk with respect to $n$ samples

$$R_n(\theta) = \frac{1}{n} \sum_{i=1}^n \ell(h(x_i; \theta), y_i) \triangleq \frac{1}{n} \sum_{i=1}^n l_i(\theta) \tag{3}$$

where $l_i(\theta) \triangleq \ell(h(x_i; \theta), y_i)$. Under stochastic programming, Eq.3 can be cast as

$$R_n(\theta) = \mathbb{E}_\omega[l_\omega(\theta)] \tag{4}$$

where $\omega \sim \text{Uniform}\{1, \cdots, n\}$. In the conventional SGD setting, at iteration $k$, a sample $(x_{i_k}, y_{i_k})$, $i_k \in \{1, \cdots, n\}$, is drawn at random and the stochastic gradient $\nabla l_{i_k}$ is then used to update $\theta$ with an appropriate stepsize $\alpha_k > 0$:

$$\theta_{k+1} = \theta_k - \alpha_k \nabla l_{i_k}(\theta_k). \tag{5}$$

In conventional SGD optimization of Eq.3 or Eq.4, there is only one parameter vector $\theta$ under consideration. We further assume $\theta$ follows some distribution $p(\theta)$ and consider the expected empirical risk over $p(\theta)$

$$J = \mathbb{E}_\theta[R_n(\theta)] = \mathbb{E}_\theta[\mathbb{E}_\omega[l_\omega(\theta)]] \tag{6}$$

In practice, a population of $\mu$ candidate solutions, $\{\theta_j\}_{j=1}^\mu$, is drawn and we deal with the following average empirical risk of the population

$$J_\mu = \frac{1}{\mu} \sum_{j=1}^\mu R_n(\theta_j) = \frac{1}{\mu} \sum_{j=1}^\mu \left( \frac{1}{n} \sum_{i=1}^n l_i(\theta_j) \right) \tag{7}$$

Eq.6 and Eq.7 formulate the objective function of the proposed ESGD algorithm. Following the EA terminology, we interpret the empirical risk $R_n(\theta)$ given parameter $\theta$ as the fitness function of $\theta$ which we want to minimize. [1] We want to choose a population of parameter $\theta$, $\{\theta_j\}_{j=1}^\mu$, such that the whole population or its selected subset has the best average fitness values.

**Definition 1** (*$m$-elitist average fitness*). Let $\Psi_\mu = \{\theta_1, \cdots, \theta_\mu\}$ be a population with $\mu$ individuals $\theta_j$ and let $f$ be a fitness function associated with each individual in the population. Rank the individuals in the ascending order

$$f(\theta_{1:\mu}) \le f(\theta_{2:\mu}) \le \cdots \le f(\theta_{\mu:\mu}) \tag{8}$$

where $\theta_{k:\mu}$ denotes the $k$-th best individual of the population [9]. The **$m$-elitist average fitness** of $\Psi_\mu$ is defined to be the average of fitness of the first $m$-best individuals ($1 \le m \le \mu$)

$$J_{\bar{m}:\mu} = \frac{1}{m} \sum_{k=1}^m f(\theta_{k:\mu}) \tag{9}$$

Note that, when $m = \mu$, $J_{\bar{m}:\mu}$ amounts to the average fitness of the whole population. When $m = 1$, $J_{\bar{m}:\mu} = f(\theta_{1:\mu})$, the fitness of the single best individual of the population.

## 3.2 Algorithm

ESGD iteratively optimizes the $m$-elitist average fitness of the population defined in Eq.9. The evolution inside each ESGD generation for improving $J_{\bar{m}:\mu}$ alternates between the SGD step, where each individual $\theta_j$ is updated using the stochastic gradient of the fitness function $R_n(\theta_j)$, and the evolution step, where the gradient-free EA is applied using certain transformation and selection operators. The overall procedure is given in Algorithm 1.

To initialize ESGD, a parent population $\Psi_\mu$ with $\mu$ individuals is first created. This population evolves in generations. Each generation consists of an SGD step followed by an evolution step. In the SGD

**Algorithm 1:** Evolutionary Stochastic Gradient Descent (ESGD)

---

**Input:** generations $K$, SGD steps $K_s$, evolution steps $K_v$, parent population size $\mu$, offspring population size $\lambda$ and elitist level $m$.

Initialize population $\Psi_\mu^{(0)} \leftarrow \{\theta_1^{(0)}, \cdots, \theta_\mu^{(0)}\}$;

// $K$ generations

**for** $k = 1 : K$ **do**

    Update population $\Psi_\mu^{(k)} \leftarrow \Psi_\mu^{(k-1)}$;

    // in parallel

    **for** $j = 1 : \mu$ **do**

        Pick an optimizer $\pi_j^{(k)}$ for individual $\theta_j^{(k)}$;

        Select hyper-parameters of $\pi_j^{(k)}$ and set a learning schedule;

        // $K_s$ SGD steps

        **for** $s = 1 : K_s$ **do**

            SGD update of individual $\theta_j^{(k)}$ using $\pi_j^{(k)}$;

            If the fitness degrades, the individual backs off to the previous step $s-1$.

        **end**

    **end**

    // $K_v$ evolution steps

    **for** $v = 1 : K_v$ **do**

        Generate offspring population $\Psi_\lambda^{(k)} \leftarrow \{\theta_1^{(k)}, \cdots, \theta_\lambda^{(k)}\}$;

        Sort the fitness of the parent and offspring population $\Psi_{\mu+\lambda}^{(k)} \leftarrow \Psi_\mu^{(k)} \bigcup \Psi_\lambda^{(k)}$ ;

        Select the top $m$ ($m \leq \mu$) individuals with the best fitness ($m$-elitist);

        Update population $\Psi_\mu^{(k)}$ by combining $m$-elitist and randomly selected $\mu-m$ non-$m$-elitist candidates;

    **end**

**end**

---

step, an SGD-based optimizer $\pi_j$ with certain hyper-parameters and learning schedule is selected for each individual $\theta_j$ which is updated by $K_s$ epochs. In this step, there is no interaction between the optimizers. From the EA perspective, their gene isolation as a species is preserved. After each epoch, if the individual has a degraded fitness, $\theta_j$ will back off to the previous epoch. After the SGD step, the gradient-free evolution step follows. In this step, individuals in the parent population $\Psi_\mu$ start interacting via model combination and mutation to produce an offspring population $\Psi_\lambda$ with $\lambda$ offsprings. An $m$-elitist strategy is applied to the combined population $\Psi_{\mu+\lambda} = \Psi_\mu \bigcup \Psi_\lambda$ where the $m$ ($m \leq \mu$) individuals with the best fitness are selected, together with the rest $\mu-m$ randomly selected individuals to form the new parent population $\Psi_\mu$ for the next generation.

The following theorem shows that the proposed ESGD given in Algorithm 1 guarantees that the $m$-elitist average fitness will never degrade.

**Theorem 1.** *Let $\Psi_\mu$ be a population with $\mu$ individuals $\{\theta_j\}_{j=1}^\mu$. Suppose $\Psi_\mu$ evolves according to the ESGD algorithm given in Algorithm 1 with back-off and $m$-elitist. Then for each generation $k$,*

$$J_{\bar{m}:\mu}^{(k)} \leq J_{\bar{m}:\mu}^{(k-1)}, \quad k \geq 1$$

The proof of the theorem is given in the supplementary material. From the theorem, we also have the following corollary regarding the $m$-elitist average fitness.

**Corollary 1.** $\forall m'$, $1 \leq m' \leq m$, *we have*

$$J_{\bar{m}':\mu}^{(k)} \leq J_{\bar{m}':\mu}^{(k-1)}, \text{ for } k \geq 1. \tag{10}$$

*Particularly, for $m' = 1$, we have*

$$f^{(k)}(\theta_{1:\mu}) \leq f^{(k-1)}(\theta_{1:\mu}), \text{ for } k \geq 1. \tag{11}$$

*The fitness of the best individual in the population never degrades.*

## 3.3 Implementation

In this section, we give the implementation details of ESGD. The initial population is created either by randomized initialization of the weights of networks or by perturbing some existing networks. In the SGD step of each generation, a family of SGD-based optimizers (e.g. conventional SGD and ADAM) is considered. For each selected optimizer, a set of hyper-parameters (e.g. learning rate, momentum, Nesterov acceleration and dropout rate) is chosen and a learning schedule is set. The hyper-parameters are randomly selected from a pre-defined range. In particular, an annealing schedule is applied to the range of the learning rate over generations.

In the evolution step there are a wide variety of evolutionary algorithms that can be considered. Despite following similar biological principles, these algorithms have diverse evolving diagrams. In this work, we use the $(\mu/\rho+\lambda)$-ES [6]. Specifically, we have the following transformation and selection schemes:

1. Encoding: Parameters are vectorized into a real-valued vector in the continuous space.
2. Selection, recombination and mutation: In generation $k$, $\rho$ individuals are selected from the parent population $\Psi_\mu^{(k)}$ using roulette wheel selection where the probability of selection is proportional to the fitness of an individual [7]. An individual with better fitness has a higher probability to be selected. $\lambda$ offsprings are generated to form the offspring population $\Psi_\lambda^{(k)}$ by intermediate recombination followed by a perturbation with the zero-mean Gaussian noise, which is given in Eq.12.

$$\theta_i^{(k)} = \frac{1}{\rho}\sum_{j=1}^{\rho}\theta_j^{(k)} + \epsilon_i^{(k)} \tag{12}$$

   where $\theta_i \in \Psi_\lambda^{(k)}$, $\theta_j \in \Psi_\mu^{(k)}$ and $\epsilon_i^{(k)} \sim \mathcal{N}(0, \sigma_k^2)$. An annealing schedule may be applied to the mutation strength $\sigma_k^2$ over generations.
3. Fitness evaluation: After the offspring population is generated, the fitness value for each individual in $\Psi_{\mu+\lambda}^{(k)} = \Psi_\mu^{(k)}\bigcup\Psi_\lambda^{(k)}$ is evaluated.
4. $m$-elitist: $m$ $(1\leq m\leq\mu)$ individuals with the best fitness are first selected from $\Psi_{\mu+\lambda}^{(k)}$. The rest $\mu-m$ individuals are then randomly selected from the other $\mu+\lambda-m$ candidates in $\Psi_{\mu+\lambda}^{(k)}$ to form the parent population $\Psi_\mu^{(k+1)}$ of the next generation.

After ESGD training is finished, the candidate with the best fitness in the population $\theta_{1:\mu}$ is used as the final model for classification or regression tasks. All the SGD updates and fitness evaluation are carried out in parallel on a set of GPUs.

## 4 Experiments

We evaluate the performance of the proposed ESGD on large vocabulary continuous speech recognition (LVCSR), image recognition and language modeling. We compare ESGD with two baseline systems. The first baseline system, denoted "single baseline" when reporting experimental results, is a well-established single model system with respect to the application under investigation, trained using certain SGD-based optimizer with appropriately selected hyper-parameters following certain training schedule. The second baseline system, denoted "population baseline", is a population-based system with the same size of population as ESGD. Optimizers being considered are SGD and ADAM except in image recognition where only SGD variants are considered. The optimizers together with their hyper-parameters are randomly decided at the beginning and then fixed for the rest of the training with a pre-determined training schedule. This baseline system is used to mimic the typical hyper-parameter tuning process when training deep neural network models. We also conducted ablation experiments where the evolution step is removed from ESGD to investigate the impact of evolution. The $m$-elitist strategy is applied to 60% of the parent population.

### 4.1 Speech Recognition

**BN50** The 50-hour Broadcast News is a widely used dataset for speech recognition [31]. The 50-hour data consists of 45-hour training set and a 5-hour validation set. The test set comprises 3

hours of audio from 6 broadcasts. The acoustic models we used in the experiments are fully-connected feed-forward network with 6 hidden layers and one softmax output layer with 5,000 states. There are 1,024 units in the first 5 hidden layers and 512 units in the last hidden layer. Sigmoid activation functions are used for all hidden units except the bottom 3 hidden layers in which ReLU functions are used. The fundamental acoustic features are 13-dimensional Perceptual Linear Predictive (PLP) [32] coefficients. The input to the network is 9 consecutive 40-dimensional speaker-adapted PLP features after linear discriminative analysis (LDA) projection from adjacent frames.

**SWB300** The 300-hour Switchboard dataset is another widely used dataset in speech recognition [31]. The test set is the Hub5 2000 evaluation set composed of two parts: 2.1 hours of switchboard (SWB) data from 40 speakers and 1.6 hours of call-home (CH) data from 40 different speakers. Acoustic models are bi-directional long short-term memory (LSTM [33]) networks with 4 LSTM layers. Each layer contains 1,024 cells with 512 in each direction. On top of the LSTM layers, there is a linear bottleneck layer with 256 hidden units followed by a softmax output layer with 32,000 units. The LSTMs are unrolled 21 frames. The input dimensionality is 140 which comprises 40-dimensional speaker-adapted PLP features after LDA projection and 100-dimensional speaker embedding vectors (i-vectors [34]).

The networks are optimized under the cross-entropy criterion. The single baseline is trained using SGD with a batch size 128 without momentum for 20 epochs. The initial learning rate is 0.001 for BN50 and 0.025 for SWB300. The learning rate is annealed by 2x every time the loss on the validation set of the current epoch is worse than the previous epoch and meanwhile the model is backed off to the previous epoch. The population sizes for both baseline and ESGD are 100. The offspring population of ESGD consists of 400 individuals. In ESGD, after 15 generations ($K_s = 1$), a 5-epoch fine-tuning is applied to each individual with a small learning rate. The details of experimental configuration are given in the supplementary material.

Table 1 shows the results of two baselines and ESGD on BN50 and SWB300, respectively. Both the validation losses and word error rates (WERs) are presented for the best individual and the top 15 individuals of the population. For the top 15 individuals, a range of losses and WERs are presented. From the tables, it can be observed that the best individual of the population in ESGD significantly improves the losses and also improves the WERs over both the single baseline as well as the population baseline. Note that the model with the best loss may not give the best WER in some cases, although typically they correlate well. The ablation experiment shows that the interaction between individuals in the evolution step of ESGD is helpful, and removing the evolution step hurts the performance in both cases.

## 4.2 Image Recognition

The CIFAR10 [35] dataset is a widely used image recognition benchmark. It contains a 50K image training-set and a 10K image test-set. Each image is a 32x32 3-channel color image. The model used in this paper is a depth-20 ResNet model [36] with a 64x10 linear layer in the end. The ResNet is trained under the cross-entropy criterion with batch-normalization. Note that CIFAR10 does not include a validation set. To be consistent with the training set used in the literature, we do not split a validation-set from the training-set. Instead, we evaluate training fitness over the entire training-set. For the single-run baseline, we follow the recipes proposed in [37], in which the initial learning rate is 0.1 and gets annealed by 10x after 81 epochs and then annealed by another 10x at epoch 122. Training finishes in 160 epochs. The model is trained by SGD using Nesterov acceleration with a momentum 0.9. The classification error rate of the single baseline is 8.34%. In practice, we found for this workload, ESGD works best when only considering SGD optimizer with randomized hyper-parameters (e.g., learning rate and momentum). We record the detailed experimental configuration in the supplementary material. The CIFAR10 results in Table 1 indicate that ESGD clearly outperforms the two baselines in both training fitness and classification error rates.

## 4.3 Language Modeling

The evaluation of the ESGD algorithm is also carried out on the standard language modeling task Penn Treebank (PTB) dateset [38]. Hyper-parameters have been massively optimized over the previous years. The current state-of-the-art results are reported in [39] and [40]. Hence, our focus is to investigate the effect of ESGD on top of a state-of-the-art 1-layer LSTM LM training recipe [40].

The results are summarized in Table 1. Starting from scratch, the single baseline model converged after 574 epochs and achieved a perplexity of 67.3 and 64.6 on the validation and evaluation sets. The population baseline models are initialized by cloning the single baseline and generating offsprings by mutation. Then optimizers (SGD and ADAM) are randomly picked and models are trained for 300 epochs. Randomizing the optimizer includes additional hyper-parameters like the various dropout ratios/models ([41, 42, 43, 40, 44, 45]), batch size, etc. For comparison, the warm restart of the single baseline gives 0.2 perplexity improvement on the test set [46]. Using ESGD, we also relax the back-off to the initial model by probability of $p_{\text{backoff}} = 0.7$ in each generation. The single baseline model is always added to each generation without any update, which guarantees that the population can not perform worse than the single baseline. The detailed parameter settings are provided in the supplementary material. ESGD without the evolutionary step clearly shows difficulties, and the best model's "gene" can not become prevalent in the successive generations according to the proportion of its fitness value. In summary, we observe small but consistent gain by fine-tuning existing, highly optimized model with ESGD.

Note that the above implementation for PTB experiments can be viewed as another variant of ESGD: Suppose we have a well-trained model (e.g. a competitive baseline model) which is always inserted into the population in each generation of evolution. The $m$-elitist strategy will guarantee that the best model in the population is not worse than this well-trained model even if we relax the back-off in SGD with probability.

In Fig.1 we show the fitness as a function of ESGD generations in the four investigated tasks.

Table 1: Performance of single baseline, population baseline and ESGD on BN50, SWB300, CIFAR10 and PTB. For ESGD, the tables show the losses and classification error rates of the best individual as well as the top 15 individuals in the population for the first three tasks. In PTB, the perplexities (ppl), which is the exponent of loss, of the validation set and test set are presented. The tables also present the results of the ablation experiments where the evolution step is removed from ESGD.

| BN50 | loss | | WER | |
|---|---|---|---|---|
| | $\theta_{1:\mu}$ | $\theta_{1:\mu} \leftrightarrow \theta_{15:\mu}$ | $\theta_{1:\mu}$ | $\theta_{1:\mu} \leftrightarrow \theta_{15:\mu}$ |
| single baseline | 2.082 | – | 17.4 | – |
| population baseline | 2.029 | [2.029, 2.062] | 17.1 | [16.9, 17.6] |
| ESGD w/o evolution | 2.036 | [2.036, 2.075] | 17.4 | [17.1, 17.7] |
| ESGD | 1.916 | [1.916, 1.920] | 16.4 | [16.2, 16.4] |

| SWB300 | loss | | SWB WER | | CH WER | |
|---|---|---|---|---|---|---|
| | $\theta_{1:\mu}$ | $\theta_{1:\mu} \leftrightarrow \theta_{15:\mu}$ | $\theta_{1:\mu}$ | $\theta_{1:\mu} \leftrightarrow \theta_{15:\mu}$ | $\theta_{1:\mu}$ | $\theta_{1:\mu} \leftrightarrow \theta_{15:\mu}$ |
| single baseline | 1.648 | – | 10.4 | – | 18.5 | – |
| population baseline | 1.645 | [1.645, 1.666] | 10.4 | [10.3, 10.7] | 18.2 | [18.2, 18.8] |
| ESGD w/o evolution | 1.626 | [1.626, 1.641] | 10.3 | [10.3, 10.7] | 18.3 | [18.0, 18.6] |
| ESGD | 1.551 | [1.551, 1.557] | 10.0 | [10.0, 10.1] | 18.2 | [18.0, 18.3] |

| CIFAR10 | loss | | error rate | |
|---|---|---|---|---|
| | $\theta_{1:\mu}$ | $\theta_{1:\mu} \leftrightarrow \theta_{15:\mu}$ | $\theta_{1:\mu}$ | $\theta_{1:\mu} \leftrightarrow \theta_{15:\mu}$ |
| single baseline | 0.0176 | – | 8.34 | – |
| population baseline | 0.0151 | [0.0151, 0.0164] | 8.24 | [7.90, 8.69] |
| ESGD w/o evolution | 0.0147 | [0.0147, 0.0166] | 8.49 | [7.86, 8.53] |
| ESGD | 0.0142 | [0.0142, 0.0159] | 7.52 | [7.43, 8.10] |

| PTB | validation ppl | | test ppl | |
|---|---|---|---|---|
| | $\theta_{1:\mu}$ | $\theta_{1:\mu} \leftrightarrow \theta_{15:\mu}$ | $\theta_{1:\mu}$ | $\theta_{1:\mu} \leftrightarrow \theta_{15:\mu}$ |
| single baseline | 67.27 | – | 64.58 | – |
| population baseline | 66.58 | [66.58, 68.04] | 63.96 | [63.96, 64.58] |
| ESGD w/o evolution | 67.27 | [67.27, 79.25] | 64.58 | [64.58, 76.64] |
| ESGD | 66.29 | [66.29, 66.30] | 63.73 | [63.72, 63.74] |

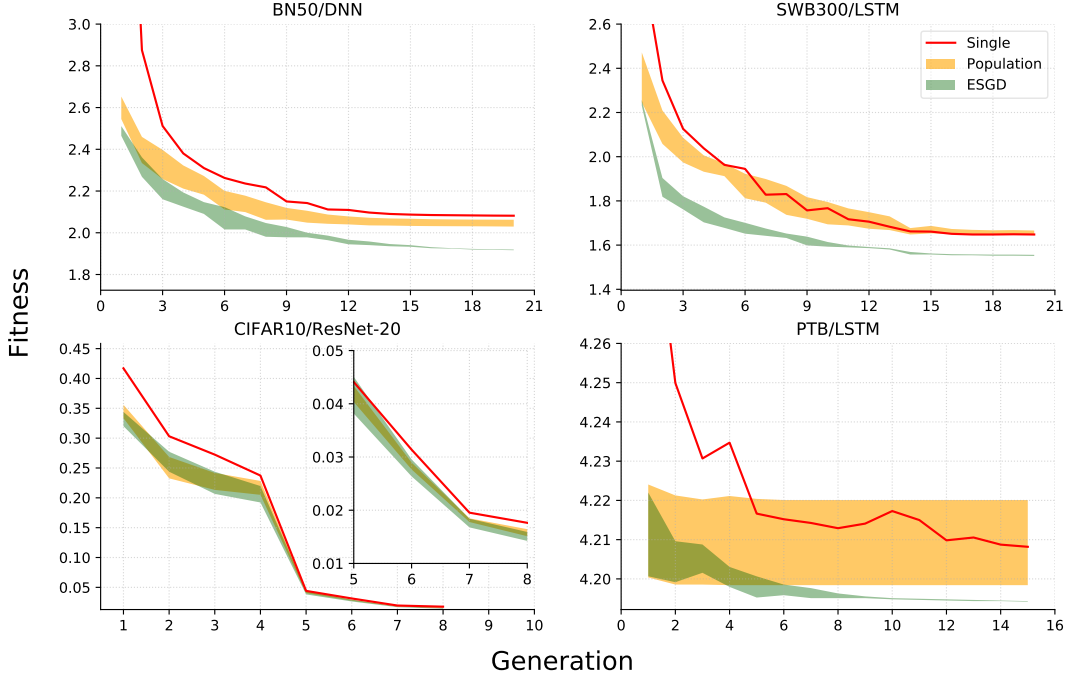

Figure 1: Fitness as a function of ESGD generations for BN50, SWB300, CIFAR10 and PTB. The three curves represent the single baseline (red), top 15 individuals of population baseline (orange) and ESGD (green). The latter two are illustrated as bands. The lower bounds of the ESGD curve bands indicate the best fitness values in the populations which are always non-increasing. Note that in the PTB case, this monotonicity is violated since the back-off strategy was relaxed with probability, which explains the increase of perplexity in some generations.

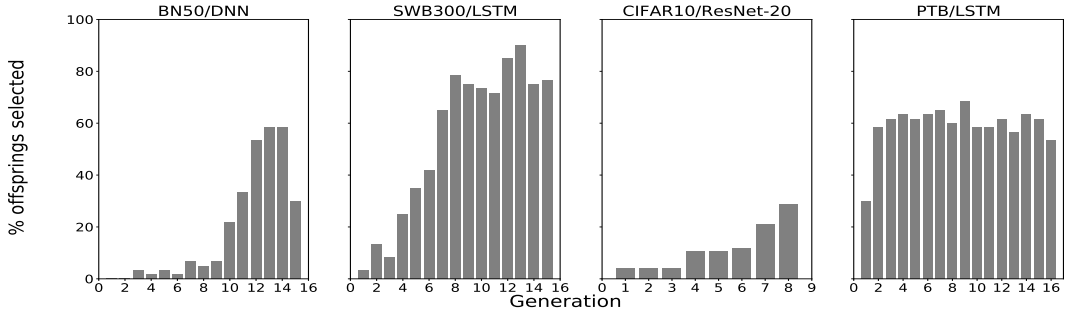

Figure 2: Percentage of offsprings selected in the 60% m-elitist over generations of ESGD in BN50, SWB300, CIFAR10 and PTB.

## 4.4 Discussion

**Population diversity** It is important to maintain a good population diversity in EA to avoid premature convergence due to the homogeneous fitness among individuals. In experiments, we find that the $m$-elitist strategy applied to the whole population, although has a better overall average fitness, can give rise to premature convergence in the early stage. Therefore, we set the percentage of $m$-elitist to 60% of the population and the remaining 40% of the population is generated by random selection. This $m$-elitist strategy is helpful in practice.

**Population evolvement** The SGD step of ESGD mimics the coevolution mechanism between competing species (individuals under different optimizers) where distinct species evolute independently. The evolution step of ESGD allows the species to interact with each other to hopefully produce promising candidate solutions for the next generation. Fig.2 shows the percentage of offsprings selected in the 60% $m$-elitist for the next generation. From the table, in the early stage the population

evolves dominantly based on SGD since the offsprings are worse than almost all the parents. However, in late generations the number of elite offsprings increases. The interaction between distinct optimizers starts to play an important role in selecting better candidate solutions.

**Complementary optimizers** In each generation of ESGD, an individual selects an optimizer from a pool of optimizers with certain hyper-parameters. In most of the experiments, the pool of optimizers consists of SGD variants and ADAM. It is often observed that ADAM tends to be aggressive in the early stage but plateaus quickly. SGD, however, starts slow but can get to better local optima. ESGD can automatically choose optimizers and their appropriate hyper-parameters based on the fitness value during the evolution process so that the merits of both SGD and ADAM can be combined to seek a better local optimal solution to the problem of interest. In the supplementary material examples are given where we show the optimizers with their training hyper-parameters selected by the best individuals in ESGD in each generation. It indicates that over generations different optimizers are automatically chosen by ESGD giving rise to a better fitness value.

**Parallel computation** In the experiments of this paper, all SGD updates and EA fitness evaluations are carried out in parallel using multiple GPUs. The SGD updates dominate the ESGD computation. The EA updates and fitness evaluations have a fairly small computational cost compared to the SGD updates. Given sufficient computing resource (e.g. $\mu$ GPUs), ESGD should take about the same amount of time as one end-to-end vanilla SGD run. Practically, trade-off has to be made between the training time and performance under the constraint of computational budget. In general, parallel computation is suitable and preferred for population-based optimization.

## 5    Conclusion

We have presented the population-based ESGD as an optimization framework to combine SGD and gradient-free evolutionary algorithm to explore their complementarity. ESGD alternately optimizes the $m$-elitist average fitness of the population between an SGD step and an evolution step. The SGD step can be interpreted as a coevolution mechanism where individuals under distinct optimizers evolve independently and then interact with each other in the evolution step to hopefully create promising candidate solutions for the next generation. With an appropriate decision strategy, the fitness of the best individual in the population is guaranteed to be non-degrading. Extensive experiments have been carried out in three applications using various neural networks with deep architectures. The experimental results have demonstrated the effectiveness of ESGD.

## Footnotes

[1]Conventionally one wants to increase the fitness. But to keep the notation uncluttered we will define the fitness function here as the risk function which we want to minimize.

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
