[Supplementary Material]

# Supplementary Material: "Evolutionary Stochastic Gradient Descent for Optimization of Deep Neural Networks"

**Xiaodong Cui, Wei Zhang, Zoltán Tüske and Michael Picheny**
IBM Research AI
IBM T. J. Watson Research Center
Yorktown Heights, NY 10598, USA
{cuix, weiz, picheny}@us.ibm.com, {Zoltan.Tuske}@ibm.com

**Theorem 1.** *Let $\Psi_\mu$ be a population with $\mu$ individuals $\{\theta_j\}_{j=1}^\mu$. Suppose $\Psi_\mu$ evolves according to the ESGD algorithm given in Algorithm 1 with back-off and $m$-elitist. Then for each generation $k$,*

$$J_{\bar{m}:\mu}^{(k)} \le J_{\bar{m}:\mu}^{(k-1)}, \quad k \ge 1$$

*Proof.* Given the parent population at the $k$-th generation, $\Psi_\mu^{(k)} = \{\theta_1^{(k)}, \cdots, \theta_\mu^{(k)}\}$, consider the SGD step of ESGD. Let $\theta_{j,s}^{(k)}$ denote the individual $j$ after epoch $s$. If

$$R_n(\theta_{j,s}^{(k)}) > R_n(\theta_{j,s-1}^{(k)}), \quad s \ge 1$$

the individual will back off to the previous update

$$\theta_{j,s}^{(k)} = \theta_{j,s-1}^{(k)}$$

Thus we have

$$R_n(\theta_{j,K_s}^{(k)}) \le R_n(\theta_{j,0}^{(k)})$$

Let

$$\tilde{J}_{\bar{m}:\mu,s}^{(k)} = \frac{1}{m} \sum_{j=1}^m R(\theta_{j:\mu,s}^{(k)})$$

be the $m$-elitist average fitness of the population after $s$ SGD epochs in generation $k$. It follows that after the SGD step,

$$\tilde{J}_{\bar{m}:\mu,K_s}^{(k)} \le \tilde{J}_{\bar{m}:\mu,0}^{(k)} = J_{\bar{m}:\mu}^{(k-1)} \tag{1}$$

where the equation

$$\tilde{J}_{\bar{m}:\mu,0}^{(k)} = J_{\bar{m}:\mu}^{(k-1)} \tag{2}$$

is due to the fact that the starting parent population of SGD of generation $k$ is the population after the generation $k-1$.

Now consider the evolution step, since the $m$-elitist is conducted on $\Psi_{\mu+\lambda}^{(k)} = \Psi_\mu^{(k)} \bigcup \Psi_\lambda^{(k)}$, it is obvious that

$$J_{\bar{m}:\mu}^{(k)} \le \tilde{J}_{\bar{m}:\mu,K_s}^{(k)} \tag{3}$$

where $J_{\bar{m}:\mu}^{(k)}$ is the $m$-elitist average fitness after the evolution step in generation $k$.

From Eq.1 and Eq.3, we have

$$J_{\bar{m}:\mu}^{(k)} \le J_{\bar{m}:\mu}^{(k-1)}, \text{ for } k \ge 1 \tag{4}$$

This completes the proof. □

# 1 Details on speech recognition experiments

The optimizers being considered for ESGD are SGD and ADAM. For SGD, randomized hyper-parameters are learning rate, momentum and nesterov acceleration. Specifically, there is a 20% chance not using momentum and 80% chance using it. When using the momentum, there is a 50% chance using the nesterov acceleration. The momentum is randomly selected from $[0.1, 0.9]$. The learning rate is also randomly selected from a range $[a_k, b_k]$ depending on the generation $k$. The upper and lower bounds of the range are annealed over generations starting from the initial range $[a_0, b_0]$. For ADAM, $\beta_1 = 0.9$ and $\beta_2 = 0.999$ are fixed and only the learning rate is randomized. The selection strategy of learning rate is analogous to that of SGD except starting with a distinct initial range. For ESGD, the mutation strength $\sigma_k$ (i.e. the variance of the Gaussian noise perturbation) is also annealed over generations. Tables 1 and 2 give the hyper-parameter settings for the ESGD experiments on BN50 and SWB300.

Table 1: Hyper-parameters of ESGD for BN50

|  | SGD | ADAM |
|---|---|---|
| $\mu$ | 100 | 100 |
| $\lambda$ | 400 | 400 |
| $\rho$ | 2 | 2 |
| $K_s$ | 1 | 1 |
| $K_v$ | 1 | 1 |
| $a_0$ | 1e-4 | 1e-4 |
| $b_0$ | 2e-3 | 1e-3 |
| $\gamma$ | 0.9 | 0.9 |
| $a_k$ | $\gamma^k a_0$ | $\gamma^k a_0$ |
| $b_k$ | $\gamma^k b_0$ | $\gamma^k b_0$ |
| momentum | $[0.1, 0.9]$ | $[0.1, 0.9]$ |
| $\sigma_0$ | 0.01 | 0.01 |
| $\sigma_k$ | $\frac{1}{k}\sigma_0$ | $\frac{1}{k}\sigma_0$ |

Table 2: Hyper-parameters of ESGD for SWB300

|  | SGD | ADAM |
|---|---|---|
| $\mu$ | 100 | 100 |
| $\lambda$ | 400 | 400 |
| $\rho$ | 2 | 2 |
| $K_s$ | 1 | 1 |
| $K_v$ | 1 | 1 |
| $a_0$ | 1e-2 | 5e-5 |
| $b_0$ | 3e-2 | 1e-3 |
| $\gamma$ | 0.9 | 0.9 |
| $a_k$ | $\gamma^k a_0$ | $\gamma^k a_0$ |
| $b_k$ | $\gamma^k b_0$ | $\gamma^k b_0$ |
| momentum | $[0.1, 0.9]$ | $[0.1, 0.9]$ |
| $\sigma_0$ | 0.01 | 0.01 |
| $\sigma_k$ | $\frac{1}{k}\sigma_0$ | $\frac{1}{k}\sigma_0$ |

# 2 Details on image recognition experiments

In the CIFAR10 experiment, training-data transformation takes three steps: (1) color normalization, (2) horizontal flip, and (3) random cropping. Test-data transformation is done via color normalization.

Initially, we mixed ADAM optimizer and SGD optimizer with a 20/80 ratio. While this setup enabled ESGD to yield significantly better evaluation loss, it however generated less accurate models due to the apparent Adam's poor generalization on CIFAR dataset. This phenomenon is consistent with

what is reported in [1, 2] that adaptive gradient methods, such like ADAM, suffer from generalization problem on CIFAR10. We also experimented with different probability of turning on Nesterov momentum acceleration, setting different momentum range, yet we did not observe noticeable improvement over baseline. Eventually, we found the configuration that consistently works the best is the following setup: Multiply base learning rate [1] with a number randomly sampled between 0.9 and 1.1. Always turn on Nesterov momentum and set it to be 0.9. When SGD yields a worse model (in terms of fitness score), always roll back to the model from the previous generation. The elitist parameter $m$ is set to be 60% of the population and number of parents is 2. The mutation strength is set to be $\frac{1}{k} \times 0.01$, where $k$ is the generation number. We summarize this setup in Table 3.

Table 3: Hyper-parameters of ESGD for CIFAR10

|  | SGD |
| --- | --- |
| $\mu$ | 128 |
| $\lambda$ | 768 |
| $\rho$ | 2 |
| $K_s$ | 20 |
| $K_v$ | 1 |
| $a_k$ | 0.9 |
| $b_k$ | 1.1 |
| momentum | 0.9 |
| $\sigma_0$ | 0.01 |
| $\sigma_k$ | $\frac{1}{k}\sigma_0$ |

## 3    Details on language modeling experiments

The Penn Treebank dataset is defined as a subsection of the Wall Street Journal corpus. Its preprocessed version contains roughly one million running words, a vocabulary of 10k words, and well defined training, validation, and test sets. Our model is based on [3]: the embedding layer of the model contained 655 nodes, and the size of a single LSTM hidden layer is equal to the embedding size. The model uses various dropout techniques: hidden activation, weight/embedding, variational, fraternal dropout [3, 4, 5, 6]. In contrast to the other experimental setups, picking an optimizer also includes the randomization of the following hyper-parameters during population based training (ESGD or population baseline): embedding, hidden activation, LSTM weight dropout ratios, mini batch size, patience, weight decay, dropout model (expectation linear dropout (ELD), fraternal dropout (FD), $\Pi$- (PD) or standard single output model). Population size, number of offsprings, SGD variant and ADAM optimizer, momentum, and learning rate intervals were chosen similar to the ASR setup. The population size was fixed to 100, elitism to 60% of the population and 400 offsprings were generated before the selection step. The ESGD ran for 15 generations and each SGD step performed 20 epochs before carrying out the evolutionary step. Furthermore, in case of unsuccessful SGD step, the backing off was deactivated by 0.3 probability. The parameters and their range are shown in Table 4. The first population was initialized by mutation of baseline model. In addition, the baseline was specifically added to the population before the evolutionary step.

## 4    Examples on complementary optimizers

Table 5 gives the selected optimizers together with their hyper-parameters for each generation of ESGD in BN50 and SWB300.

Table 4: Hyper-parameters of ESGD for Penn Treebank

| | SGD | ADAM |
|---|---|---|
| $\mu$ | | 100 |
| $\lambda$ | | 400 |
| $\rho$ | | {1,2} |
| $K_s$ | | 20 |
| $K_v$ | | 1 |
| $a_0$ | 1 | 1e-6 |
| $b_0$ | 60 | 1e-3 |
| $\gamma$ | | 0.33 |
| $a_k$ | | $\gamma^k a_0$ |
| $b_k$ | | $\gamma^k b_0$ |
| $p_{\text{momentum}}$ | | 0.8 |
| $p_{\text{Nesterov}}$ | 0.5 | 0 |
| momentum | | [0.1, 0.9] |
| $\sigma_0$ | | [0.01, 0.15] |
| $\sigma_k$ | $\frac{1}{k}\sigma_0$ | $\frac{1}{k}\sigma_0$ |
| fitness smoothing | | 0.5 |
| dropout$_{\text{embedding}}$ | | [0.05, 0.2] |
| dropout$_{\text{hidden}}$ | | [0.1, 0.65] |
| dropout$_{\text{LSTM weights}}$ | | [0.1, 0.65] |
| model$_{\text{dropout}}$ | | {FD,ELD,PM,Standard} |
| patience | | [3, 10] |
| weight decay | | [0, 1.2e-5] |
| $p_{\text{backoff}}$ | | 0.7 |
| mini batch | | [20, 64] |

Table 5: The optimizers selected by the best candidate in the population over generations in ESGD in BN50 DNNs and SWB300 LSTMs.

| generation | optimizer | |
|---|---|---|
| | BN50 DNN | SWB300 LSTM |
| 1 | adam, lr=2.68e-4 | adam, lr=5.39e-4 |
| 2 | adam, lr=1.10e-4 | adam, lr=4.61e-5 |
| 3 | adam, lr=1.87e-4 | adam, lr=9.65e-5 |
| 4 | sgd, nesterov=F, lr=3.61e-4, momentum=0 | sgd, nesterov=F, lr=9.92e-3, momentum=0.21 |
| 5 | adam, lr=9.07e-5 | sgd, nesterov=T, lr=6.60e-3, momentum=0.35 |
| 6 | adam, lr=1.04e-4 | adam, lr=4.48e-5 |
| 7 | sgd, nesterov=F, lr=5.20e-4, momentum=0 | sgd, nesterov=T, lr=5.60e-3, momentum=0.11 |
| 8 | sgd, nesterov=T, lr=1.00e-4, momentum=0.44 | sgd, nesterov=T, lr=5.15e-3, momentum=0.49 |
| 9 | adam, lr=6.45e-5 | adam, lr=3.20e-5 |
| 10 | adam, lr=7.51e-5 | adam, lr=2.05e-5 |
| 11 | adam, lr=4.51e-5 | adam, lr=1.97e-5 |
| 12 | sgd, nesterov=F, lr=4.19e-5, momentum=0 | adam, lr=2.98e-5 |
| 13 | sgd, nesterov=F, lr=6.17e-5, momentum=0.25 | adam, lr=1.94e-5 |
| 14 | sgd, nesterov=F, lr=6.73e-5, momentum=0.45 | adam, lr=1.33e-5 |
| 15 | sgd, nesterov=F, lr=1.00e-4, momentum=0 | adam, lr=1.45e-5 |

## Footnotes

[1]Base learning rate is 0.1 for the first 80 epochs, 0.01 for another 41 epochs, and 0.001 for the remaining epochs