[Reviews · NeurIPS 2018]

Reviewer 1



The paper combines ES and SGD in a complementary way, not viewing as an alternative to each other. More intuitively, in the course of optimization, the author proposes to use evolutionary strategy to make optimiser adjust at different sections (geometry) of optimisation path. Since the geometry of the loss surface can differ drastically at different locations, it is reasonable to use different gradient based optimisers with different hyper parameters at different locations. Then the main question becomes how to choose a proper optimiser at different locations. The paper proposes ES for that. In terms of writing, the paper is readable and easy to follow. Simple comments for better readability are as follow. It would be better if brief explanation about ‘roullette wheel selection’ is given to be more self-contained (in line 138). It seems that psi in lines 145~147 should have superscript (k), the similar ambiguity at the last line of the algorithm1. ***Related works The paper ‘Population Based Training of Neural Networks (PBT)’ uses the similar method as ESGD. Both uses ES but PBT aims to optimise hyperparameters of neural network while ESGD tries to optimise parameters of neural network. However, it is also possible to view that ESGD optimizes specific hyperparameters (different optimizers at different sections in the course of optimization). Moreover, it looks like that PBT can basically do the similar thing as ESGD with some minor modifications. We can try to optimise optimisers at different sections of the course of optimization as a hyperparameter using PBT. Since in PBT, the under performing ones are replaced by well performing one together with their parameters, thus evaluation and mutation of parameters in ESGD can be done similarly in PBT. It appears that this work is a special case of PBT. We hope that the author make the contribution over PBT more clearly. One suggestion is to compare PBT and ESGD to show the benefit of the specific choice of ES in ESGD. ***Back-off This concept and related theorem is one thing which can differentiate ESGD from PBT. However, back off looks like a commonly used heuristic which is just using the best value in the course of optimization. The statement of the theorem is quite trivial because it basically keeps the improved value. Moreover, it seems that this theorem does not have strong practical implication since the author uses stochastic back off in one experiment, which breaks continuous improvement. ***Robustness to the choice of ESGD hyperparameters In experiments, the configuration of a couple of ESGD hyperparameters varies for different data set. Pool of optimizers in SGD step - (in CIFAR10 - only using SGD) Back-off - in LM, stochastic back-off with probability 0.7) Even though a fair amount of engineering hyperparameters is indispensible for a well performing ES, a question is whether ESGD is robust with respect to ESGD hyperparameters or consistent ESGD hyperparameter can be recommended. Some experiment showing the influence of ESGD hyperparameters can be given. ***Using larger pool of optimizers in SGD step? Since various optimizers like AdaGrad, AdaDelta, RMSProp etc. behave differently, the geometry on which they can optimise well can be different and various optimiser can take care of various geometry. Contrast to the fact that, in CIFAR10, ESGD works better only with SGD, a naive conjecture is that the performance can be improved if various of optimizers are considered to tackle various geometry. If the author has not tried ESGD with various optimizers, it would be interesting to see the result with various optimiser. Or if it has been tried already, then the explanation or guideline to choose pool of optimiser can be beneficial. ***Including random initialisation to mutation step can improve ESGD? In the proposed algorithm, random initialisation is not used and descendants are selected from parent generation with added perturbation, so there is no chance that strong mutation can happen. Can this strong mutation, which is similar to large perturbation, improve the performance?

Reviewer 2



Update: After reading the reviews from other reviewers, and the author response, I decided to stick to my original score of 6. The author response was inline with my expectations and but was not significant enough to persuade me to improve the score to 7. That being said, I'm supportive of this work and I believe it is (marginally) good enough to be accepted into NIPS2018. I would like to see more works in the future combining the best of evolutionary computing with traditional deep learning methods as I think this will help push the field forward and escape from local optima. Original Review: Review for "Evolutionary Stochastic Gradient Descent for Optimization of Deep Neural Networks" Summary: This paper proposes a scheme to combine traditional gradient-free evolutionary algorithm with SGD. The basic idea is to have a population of mu parameters of a neural network, and for each generation, train each population K_s SGD steps, and use evolution to keep track of m best solutions and randomly mutate the next for each generation. Good points: - Idea is well motivated, and I think we as a community should explore more ways to combine the best of EA and SGD methods. The algorithm is nice and simple, and works on a variety of tasks. - Experiments were chosen well (covers a lot of ground), and thorough. I didn't expect them to choose SOTA methods for everything (though perhaps other reviewers of NIPS might), but they chose a representative respectable architecture as a baseline for CIFAR-10, PTB, etc., and I think 92-93% ish on an off-the-shelf ResNet and 63-64 perplexity is good enough as baseline methods for them to convince me that their approach can improve over SGD. - They prove that their scheme guarantees that the best fitness in the population never degrades. Criticisms (and things that might be addressed): - confusing notation (see discussion points). Lots of subscripts and lower cases, hard to follow. - you should discuss the compute requirements as a tradeoff. It seems you need to basically train a neural network mu times, so when comparing to a baseline of vanilla SGD, it might be unfair to have generation on the X axis? You also need I think mu GPUs that run in parallel - from the previous point, what if I just train mu different networks, and use them as an ensemble? If I use your ResNet architecture and train mu different ResNets with different random seeds from scratch, the ensemble might be better than ESGD (or maybe it might not be!). I think you might want to discuss this, I'm not expecting you to beat an ensemble but worth a discussion on pros/cons. Discussion points: In algorithm 1, in the "for s=1:K_s do" loop, shouldn't it be nested with a "for j:1:mu" loop as well? Since you use j as a subscript, but it will be undefined. This should be addressed. For the EA, what do you think about using other algorithms such as CMA-ES or even a quasi-gradient inspired method (like the OpenAI's ES paper [1] or PEPG [2])? Is the vanilla EA scheme all we need? If someone has a single GPU machine can they still run the experiment? Is this something that can be plugged in and made to work with a few lines of code change? Can you convince a random NIPS researcher that it will be worthwhile to use this method in practice? Rating: 6 I think this work is marginally good enough to be accepted at NIPS, since I think they did a good job with proposing a well motivated idea, with theoretical foundations to back their method up, and also ran a thorough set of experiments. But honestly it's not a top inspiring paper and I feel this method as it is might not be common place (given the points I raised above). Note: I'm willing to increase the score if I believe the concerns are addressed. Please get back to me in the rebuttal, thanks, and good luck! Here are the links to the papers: [1] https://arxiv.org/abs/1703.03864 [2] http://citeseerx.ist.psu.edu/viewdoc/download;jsessionid=A64D1AE8313A364B814998E9E245B40A?doi=10.1.1.180.7104&rep=rep1&type=pdf (can't find [2] on arxiv.)

Reviewer 3



Update: After reading the reviews from other reviewers, and the author response, I decided to stick to my original score of 6. Original Review: Authors propose to combing Evolutionary Strategies with Stochastic Gradient Descent to train Deep Neural Networks. There have several recent papers that apply ES to train DNNs and many papers applying SGD, this paper provides a novel combination of the two. A comparison with an ensemble of models is missing as a baseline. Without the code it would be really hard to replicate this paper. quality: The paper presents several experiments on speech recognition, image recognition and language model. clarity: The paper is well written and clear. originality: Marginally original, the paper is a combination of two previous ideas, although the specific combination is novel. significance: Marginal, the increase computational cost is not clear if it is worth compared with running an ensemble of models.